# Quantification of the Flavor and Taste of Gonads from the Sea Urchin *Mesocentrotus nudus* Using GC–MS and a Taste-Sensing System

**DOI:** 10.3390/s20247008

**Published:** 2020-12-08

**Authors:** Satomi Takagi, Yoichi Sato, Yuko Murata, Atsuko Kokubun, Ken Touhata, Noriko Ishida, Yukio Agatsuma

**Affiliations:** 1Laboratory of Marine Plant Ecology, Graduate School of Agricultural Science, Tohoku University, Aza-Aoba, Aramaki, Aoba, Sendai, Miyagi 980-0845, Japan; satomi.takagi@port.kobe-u.ac.jp; 2Riken Food Co., Ltd., Miyauchi, Tagajyo, Miyagi 985-0844, Japan; yoi_sato@rikenfood.co.jp; 3Nishina Center for Accelerator-Based Science, RIKEN, Hirosawa, Wako, Saitama 351-0198, Japan; 4National Research Institute of Fisheries Science, Japan Fisheries Research and Education Agency, Fukuura, Kanazawa, Yokohama, Kanagawa 236-8648, Japan; betty@affrc.go.jp (Y.M.); ktouhata@affrc.go.jp (K.T.); larry@affrc.go.jp (N.I.); 5Food Analysis Laboratory, Quality Assurance Division, RIKEN VITAMIN Co., Ltd., Aoyagi, Soka, Saitama 340-0002, Japan; ats_kokubun@rike-vita.co.jp

**Keywords:** sea urchin, gonad quality, GC–MS, taste-sensing system, *Mesocentrotus nudus*

## Abstract

Sea urchin gonads are a delicious seafood item of high commercial value. Our past studies have revealed that the gonads of the sea urchin *Mesocentrotus nudus* fed the basal frond portion of fresh *Saccharina* kelp (BS) or the sporophylls of fresh *Undaria* (SU) during May–July are of high-quality. The present study investigated the flavor and taste of BS and SU gonads in comparison with those from non-fed *M. nudus* (NF) using gas chromatography–mass spectrometry (GC–MS) and gas chromatography (GC)-sniffing techniques, and a taste-sensing system. Data of the estimated intensity of taste (EIT) were compared with assessment of gonads from *M. nudus* collected from an *Eisenia* bed (fishing ground) and a barren in July. Gonads from both BS and SU released pleasant green, sour, and fruity aromas characteristic of butyl acetate, which are here recognized essential flavor components of high-quality gonads. The gonads of BS and SU had a strong umami taste compared to those of NF, and the *Eisenia* bed and the barren. The most marketable *M. nudus* gonads were assessed to be those with green and fruity aromas from butyl acetate, sweet aroma from benzaldehyde, umami EIT > 13.8, bitterness EIT < 3.1, and without any unpleasant sulfurous odor from sulfur-containing compounds.

## 1. Introduction

Sea urchin gonads are a premium delicacy of high commercial value [1]. Fresh sea urchin gonad is an important item of Japanese “sushi” cuisine. Tokyo Metropolitan Central Wholesale Market is the largest wholesale sea urchin market in the world [2], where the average price of sea urchin gonads doubled from 7306 JPY·kg^−1^ in 2008 to 14,683 JPY·kg^−1^ in 2018 [3]. Furthermore, the increase in global fish and shellfish consumption in the last decades [4], and the initiation from 2012 of imports of live, fresh and chilled sea urchin into Europe and Oceania [5], indicate an increase in worldwide popularity of sea urchin.

No quantitative or qualitative standard for sea urchin gonad has been established. Quantitative assessments of gonad size, color, texture, and taste (free amino acid contents) have been conducted to evaluate gonad quality (e.g., McBride et al. [6]; Woods et al. [7]; Takagi et al. [8]). Gonads of large size, bright orange, or yellow color (Figure 1), medium hardness, high in content of sweet-tasting alanine and low in bitter-tasting arginine are preferred [1,9,10]. Research that has analyzed the quantity of odor-active compounds in sea urchin gonads using gas chromatography–mass spectrometry (GC–MS) and gas chromatography (GC)-sniffing techniques has shown that flavor is also an important factor to evaluate gonad quality [11,12]. However, such quantitative data alone cannot portray overall desirability and the subtle interactions of each quality trait, so sensory evaluation using a panel of tasters is important [8,10,13]. Recently, a taste-sensing system composed of artificial lipids (which emulate changes in the membrane potentials of the human tongue) has been developed to resolve subjective assessments, and the effects of physical and psychological conditions on sensory evaluation (reviewed by Kobayashi et al. [14]). Using this system, seafood tastes have been evaluated for dried bonito stock [15], nori sauce and dried sheets of the red alga *Pyropia yezoensis* [16,17], the pacific oyster *Crassostrea gigas* [18], the swimming crab *Portunus trituberculatus* [19], the bluefin tuna *Thunnus orientalis*, the yellowtail *Seriola quinqueradiata*, and the squids *Sthenoteuthis oualaniensis* and *Todarodes pacificus* [16]. Thus, this system would evaluate the taste of sea urchin gonad qualitatively.

Among edible sea urchins, *Mesocentrotus nudus* is the most expensive sea urchin in the world. More than two-thirds of the total sea urchin landings in Japan are accounted for by *M. nudus* and *Strongylocentrotus intermedius* [20]. *Mesocentrotus nudus* densely distributes on barrens but has gonads of undesirable quality [8,21,22]. Recently, short-term culture experiments have been performed in order to improve the gonad quality (size, color, texture and taste) of *M. nudus* from barrens, particularly in Northern Japan (e.g., Takagi et al. [8,10,23]). Our past study has revealed that feeding fresh *Saccharina japonica* kelp during May–July can improve the gonad quality of *M. nudus* from barrens to a level where it is more desirable than wild urchins harvested from an *Eisenia bicyclis* kelp bed (fishing ground); a study that involved sensory evaluation and quantitative measurements of gonad size, color and hardness, and free amino acid content [10]. In addition, the composition of odor-active compounds in the gonads of the cultured and wild sea urchins were distinctly different [24]. More recently, Takagi et al. [25] showed markedly high levels of sweet-tasting alanine and low levels of bitter-tasting arginine in the gonads of *M. nudus* fed with sporophylls of fresh *Undaria pinnatifida* and the basal frond portion of fresh *S. japonica*. Gonads from derived from both types of feed were evaluated as the same high-quality as those served in Tokyo sushi restaurants evaluated as two- or three-star Michelin establishments [25]. The aim of the present study is to assess analyses of GC–MS, and GC-sniffing techniques and a taste-sensing system to provide more rigorous evaluation of gonad flavor and taste, as a contribution to providing an objective standard by which to measure the quality of sea urchin gonads.

In the present study, adult *M. nudus* collected from a barren were fed the basal frond portions of fresh *S. japonica*, the sporophylls of fresh *U. pinnatifida* or no food during May–July according to Takagi et al. [25]. At the end of the rearing experiment, the odor-active compounds in gonads were analyzed by GC–MS and GC-sniffing techniques, and taste analysis by the taste-sensing system was conducted. For the taste analysis, a comparison was made with gonads from wild *M. nudus* from an *E. bicyclis* kelp bed and a barren.

## 2. Materials and Methods

### 2.1. Sea Urchin Samples

*Mesocentrotus nudus* used in the rearing experiment are from the same area as those used in the study of Takagi et al. [25]. On 1 May 2017, a total of 45 adult *M. nudus* (47–53 mm diameter) were collected by scuba dive from a barren (a non-commercial area where sea urchins are not highly regarded for the quality of their gonads) at depths of 2–3 m off Nojima Island in Shizugawa Bay (38°40′ N, 141°30′ E). The sea urchins were reared in nine 10 L tanks (5 urchins per tank) supplied with running filtered seawater at a rate of two to three tank volumes per hour. The seawater was pumped up fresh from offshore waters, filtered through sand and aerated. Tanks were cleaned every 3–4 days. The feeding experiment was conducted from 10 May to 18 July 2017. Three sea urchin treatments were designed: sea urchins fed the basal frond portion of fresh *S. japonica*, which were cut into three equal lengths from the base to apex, every 3–4 days (BS); fed the sporophylls of fresh *U. pinnatifida* every 3–4 days (SU); and no food (NF). Three tanks were assigned to each treatment. All sea urchins were reared unfed for nine days until the start of the experiment.

On 18 July, all sea urchins were removed from their tanks, dissected and the gonads from each treatment were stored together. Three replicates of ca. 3 g gonad tissue were collected from each treatment for odor-active compound analysis and stored in a polystyrene container at 4 °C until the analyses. The remaining gonads of each treatment were quickly frozen at −30 °C for the taste analysis using the artificial taste-sensing system. In addition, two groups of wild sea urchins were prepared for the taste analysis. On 14 and 23 July, respectively, ten wild *M. nudus* (ca. 50 mm diameter) were collected from the same barren as that from which sea urchins were collected for the rearing experiment (WBA); and ten were collected from an *E. bicyclis* bed at a depth of 1.4 m off Nojima Island (sea urchins from a fishing ground of regarded commercial quality and value) (WEB). After dissection, all gonads were stored together for each collection site, and a total of ca. 40 g gonad was randomly collected from each group and frozen for the taste analysis.

### 2.2. Odor-Active Volatile Compound Analysis

Odor-active volatile compounds in the gonad tissues were analyzed by GC–MS and GC-sniffing techniques within 48 h after dissection, according to Sato et al. [26]. Headspace volatile compounds were collected in a large-volume static headspace (LVSH) system (Entech 7100A series, Entech Instruments Inc., Simi Valley, CA, USA). Each gonad from a container was sealed in a 375 mL glass jar for measurement of LVSH and stored in an incubator (DK400, Yamato Scientific Co., Ltd., Tokyo, Japan) at 30 °C for 10 min. After incubation, 150 mL of headspace gas was vacuum extracted from the glass jar. The volatile organic compounds (VOCs) were desorbed by thermos desorption using a pre-concentrator (Entech 7100A series, Entech Instruments Inc., Simi Valley, CA, USA) and applied to the GC–MS system.

Quantification of the volatile compounds was performed using an Agilent 6890 series gas chromatograph (Agilent Technologies Inc., Palo Alto, CA, USA) equipped with an Agilent 5975B mass-selective detector and a sniffing port. One half of the column flow was directed to the MS system, while the other half was directed to the heated sniffing port. The GC–MS system was equipped with a DB-WAX column (60 m × 0.25 mm i.d., 0.5 µm film thickness; 122-7063, Agilent Technologies Inc.). The GC operating conditions were: injector temperature, 250 °C; helium carrier gas mean linear velocity, 20.6 cm s^−1^. Temperature program (3 steps): 40 °C for 5 min; 5 °C min^−1^ increase to 240 °C; final 5 min hold at 240 °C. Mass spectrometry was carried out in scan mode using an electron ionization voltage of 70 eV and a scan range from m/z 10 to 300 every 1.58 s. Analysis of VOCs was performed using the program Powered Pro (Wiley 10th + NIST 2014 Mass Spectral Library, Wiley-VCH, Weinheim, Germany). Each VOC was identified by a similarity search [27] using a software library (Wiley 11N17main, Wiley-VCH). When a VOC was detected in triplicate analysis, this determined the presence of the VOC in the group.

One half of the column flow was directed to a heated sniffing port (ODP2 Olfactory Detection Port, Gerstel GmbH & Co.KG, Mülheim an der Ruhr, Germany) for GC-sniffing analysis. Humidified air (50–75% relative humidity) was carried to the sniffing port at 1.02 mL min^−1^. The panelists, who are well versed in sea urchin gonad quality and share common perceptions, recorded the retention time and the related description of the aroma compounds [28,29] by writing on paper. The relative amounts of each volatile compound detected by GC-sniffing analysis were calculated based on the peak areas in the chromatograms.

### 2.3. Taste Analysis

The estimated intensity of taste (EIT) of sea urchin gonads was analyzed according to the method of Touhata et al. [16] with a slight modification. Three replicates per gonad of a group were assigned. Approximately 10 g gonads of each group were finely cut into a paste with scissors and diluted with nine volumes of distilled water. The mixture was boiled for 15 min and then filtered through a nylon net (draining bag; 6-0709-0601, Daicel FineChem Ltd., Tokyo, Japan) to obtain the sample extract. The filtrates were estimated for taste intensity with a taste-sensing system (SA402B, Intelligent Sensor Technology. Inc., Kanagawa, Japan) according to the manufacturer’s protocol. The system can measure eight kinds of taste (sourness, bitterness, astringency, umami, saltiness, bitter after-taste, astringent after-taste and umami after-taste). A reference solution was used as a tasteless sample with taste criterion set to zero. Taste criteria with EIT values > 0 were evaluated, except for saltiness and sourness, where EIT values above −6 and −13, respectively, indicate taste because the reference solution contains 30 mM KCl and 0.3 mM tartaric acid. One unit of EIT variation represents a 20% concentration difference in the standard solution, which corresponds to the discrimination threshold [30], so recognizable differences in taste among groups were evaluated according to whether or not the EIT values differ by more than one unit.

## 3. Results

### 3.1. Odor-Active Compounds

A total of 30 odors were described using the GC-sniffing technique, and 19 compounds were identified as odor-active compounds (Figure 2, Table 1, Appendix A). An “unpleasant fishy” odor at 21.30 min was from dodecane and/or ethyl toluene, and an “unpleasant sea urchin” odor at 26.62–26.66 min was from 6-methly-5-hepten-2-one and/or isopropyl benzene. There were significant differences in peak areas of xylene, styrene, 3-octanol and 2-butoxyethanol from gonads among treatments (*p* < 0.05). A larger number of compounds with pleasant aromas was detected from gonads of BS (eight compounds) and SU (eight compounds) compared to those of NF (three compounds). There were more compounds with unpleasant odors from gonads of SU (eight compounds) than BS (four compounds). Scents of benzaldehyde in the sea urchin gonads were described as “pleasant”, “sweet”, “floral”, and “fresh” aromas; and those of ethyl octanoate were described as “unpleasant”, “sea urchin-like”, and “sweet-and-sour”. Benzene, butyl acetate, ethyl octanoate, and an unidentified compound at 33.39–33.50 min were detected as odor-active compounds of the gonads of both BS and SU.

Furthermore, 2-butoxyethanol and 3-octanol were detected from gonads of BS and SU, respectively. “Pleasant”, “fresh”, and “green” aromas from ethyl acetate and “unpleasant sea urchin-like” odor from limonene were detected from gonads of BS, but not from those of other treatments. The peak area of limonene from gonads of BS was larger than that of SU. “Orange”, “fresh”, and “sweet” aromas from xylene, “pleasant sea urchin-like” aroma from 3-octanal, “green” aroma from 2-ethylhexanol and “sweet” aroma from 2-methyl-6-methylene-2,7-octadienal were detected from gonads of SU, but not from those of other treatments. “Green” and “chemical” scents from toluene were detected from gonads of BS and NF, and their peak areas were large compared to those of SU.

### 3.2. Taste Analyzed by the Taste-Sensing System

The EIT values for “sourness”, “saltiness”, and “astringent after-taste” in gonads of all sea urchin groups were below −6, −13 and 0, respectively, indicating that the sensor output was below the threshold for each of these taste parameters. There were no recognizable differences among groups for EIT values of “astringency” and “umami after-taste” (Figure 3, Appendix A). EIT “bitter after-taste” values were recognizably higher for gonads of NF than for those of BS, SU, WBA, and WEB; “bitterness” was higher for NF than for other groups; and “umami” was higher for BS and SU than for other groups. Recognizably lower “bitterness” EIT values were detected for SU gonads than for those of other groups; and “umami” EIT values were recognizably lower for WEB gonads than for those of the other group.

## 4. Discussion

### 4.1. Odor-Active Compounds

The odor-active compound composition in gonads of *M. nudus* differed among treatments. The detection of a larger number of odor-active compounds from the gonads of sea urchins fed ad libitum with pieces of kelp (BS or SU) than those from starved sea urchins (NF) is in accordance with an earlier study (Takagi et al. [24]) that reported a smaller number of odor-active compounds from WBA gonads compared to those of sea urchins fed with whole fronds of *S. japonica* (WFS). The benzaldehyde and ethyl octanoate aromas detected from gonads of sea urchins in each treatment (BS, SU and NF) indicate that the aromas from both compounds exist in gonads regardless of nutritive state. Benzaldehyde from WFS gonads was described as a “sea urchin-like” aroma [24]. The compound would be typically associated with a pleasant “sweet” sea urchin aroma [11].

Decane detected in NF gonads was described as a “green” scent. Similarly, the compound was described as “green” in WBA gonads [24], suggesting that this compound and its aroma may be present in small-sized gonads of starved sea urchins.

Butyl acetate may be an important indicator of good flavor for high-quality gonads since this substance in gonads of BS and SU in the present study was described as producing pleasant aromas “green”, “sour”, and “fruity”, which is similar to the description of aromas from WFS gonads [24]. The results from this study confirm those of previous studies [24,25], indicating that the flavor of high-quality gonads is associated with a low content of sulfur-containing compounds such as S-methyl thioacetate and bis-(methylthio)-methane (which are known to produce off-flavor and “sulfur” odor in shellfish [26,31,32,33]) and pleasant aromas of butyl acetate and benzaldehyde. An omission test is required to identify the compounds constituting the overall flavor of high-quality gonads. Some odor-active compounds, such as 2-propanol, 2-ethylhexanol, S-methyl thioacetate, and bis-(methylthio)-methane from WFS gonads [24], were not detected as odor-active compounds from gonads of BS. The carbon, nitrogen, protein, and total- and free-amino-acid composition of *S. japonica* fronds vary according to position within the frond [34,35,36]. A previous study (Takagi et al. [36]) has revealed differences in free-amino-acid composition of *M. nudus* gonads associated with feeding with different parts of the *S. japonica* frond, suggesting a correlation with differences in gonad flavor.

### 4.2. Taste Analysis

The taste of sea urchin gonads analyzed by the taste-sensing system used here showed a large variation associated with different kinds of feed. The sensor showed that the gonads of BS and SU, which were evaluated as high-quality with high marketability by a sensory test [25], had a stronger umami taste compared to gonads of regarded commercially harvested sea urchins (WEB) and those of NF and WBA without commercial value. The lack of significant differences in EIT for “astringency” and “umami after-taste” among groups suggests that these taste traits are not affected by the food source.

The taste of sea urchin gonads has been evaluated by the content of free-amino-acids, nucleotide associated compounds, and organic acids in gonads [8,10,13,25,37,38,39,40,41,42]. The high EIT values for “bitter after-taste” in NF might be affected by the content of sulfur-containing compounds [17]. Akitomi et al. [43] reported that a significant decrease in the response value of a bitterness sensor was observed when arginine was added to bitter-tasting methionine, leucine, and isoleucine. On the other hand, arginine (712 mg/100 g) content in NF gonads was significantly higher than those of BS (335 mg/100 g) and SU (257 mg/100 g); and methionine, leucine and isoleucine contents in NF gonads were not high compared with those of BS and SU [25]. Therefore, the cause of the difference in bitterness among groups remains unclear.

Umami is the taste imparted by aspartic acid, glutamate and 5′-ribonucleotides such as inosinate and guanylate (reviewed by Ninomiya [44]). The umami-tasting aspartic acid content was below the threshold value (100 mg/100 g) in gonads of BS, SU and NF [25], and of WBA and WEB [10,45]. The umami-tasting glutamic acid (62 mg/100 g) content in gonads of NF was significantly lower than that of BS (205 mg/100 g) and SU (216 mg/100 g) [25], coinciding with EIT values for umami in the present study. In contrast, the glutamic acid content in WEB gonads ranged between 125 and 160 mg/100 g during May–July [8,10,23], suggesting that glutamic acid content in WEB gonads would be higher than that of NF in the present study. However, the EIT of NF gonad umami was recognizably higher than that of WEB samples. A sensory evaluation showed that the umami score in WEB gonads is high compared with that from WBA samples, although WEB glutamic acid content was significantly lower than that from WBA [10]. Kitaoka et al. [18] suggested that the umami sensor would reflect synergistic effects of glutamic acid, inosinate, and AMP by taste evaluation of the pacific oyster *C. gigas* collected from three different localities. The observed difference in umami taste might be due to a difference in nucleotides or a combination of nucleotides and amino acids [10], explaining the umami EIT values of the present study. As 5′-inosinate and 5′-guanylate are produced by the decomposition of ATP and ribonucleic acids, respectively, when the animal and its cells are dead, the contents of these compounds change between dissection and analysis (reviewed by Kurihara [46]). Sensory evaluation is susceptible to the physical and psychological condition of panel members [14] and assembling trained panels carries a significant cost in both time and money. Past studies have shown the potential of the taste-sensing system to evaluate taste of *C. gigas* [18], the swimming crab *P. trituberculatus* [19], the bluefin tuna *T. orientalis*, the yellowtail *S. quinqueradiata*, and the squids *S. oualaniensis* and *T. pacificus* [16]. Therefore, we conclude the taste-sensing system has a high potential to reliably and economically evaluate the umami taste of sea urchin gonads.

### 4.3. Standard of High-Quality Gonad of M. nudus

Research into short-term culture trials to improve the gonad quality of *M. nudus* from barren grounds where there is no sea urchin fishery has been undertaken mostly in Japan [8,10,25,42,47] but recently also in Australia [48]. Culture experiments with adult *M. nudus* that were collected from barrens have revealed that feeding them with fresh *S. japonica* or the sporophylls of fresh *U. pinnatifida* improves gonad quality as assessed by sensory evaluation, and quantitative measurements and analyses of gonad size, color (*L*a*b** values), hardness, and free-amino-acid content [8,10,25]. Takagi et al. [10] demonstrated that gonad quality can be improved in terms of increase in size, *L** value (lightness), alanine content, enhancement of umami taste, and decrease in arginine content. In addition, the present study confirmed that a decrease in unpleasant “sulfur” odor (due to sulfur-containing compounds detected in wild sea urchins [24,26]) and increase in pleasant aromas from butyl acetate and benzaldehyde can improve gonad flavor (cf. [24]). Sensory evaluation, and quantitative measurements and analyses of gonads showed that the WEB gonads, and those of cultured sea urchins evaluated as of the same or higher quality, had the following quantitative ranges: *L** value 56.0–60.7, hardness 0.11–0.14 N, alanine content 235–595 mg/100 g, and arginine content 215–334 mg/100 g [8,10,25]. The following qualitative characteristics were noted: “green” and “fruity” aromas from butyl acetate, “sweet” aroma from benzaldehyde ([24], the present study), and bitterness EIT < 3.1 (the present study). These values are suitable as a preliminary standard for sea urchin gonads of commercial value.

Further, the results of sensory evaluation and quantitative data of BS and SU gonads indicate that those containing > 519 mg/100 g of alanine and < 255 mg/100 g of arginine [25], without an unpleasant sulfur odor, and with an umami EIT > 13.8 (present study) can be highly marketable. As *M. nudus* is the most expensive sea urchin on the world market [20], these quality standards could be applicable worldwide.

Both *S. japonica* and *U. pinnatifida* are cultivated abundantly in Japan, Korea, and China and the total production in these countries was 13,768,680 t (in wet weight) in 2018 [49]. In Japan, the basal frond portion *S. japonica* remains on the cultivation rope after the harvest and is then discarded (Ohkami, Fudai Fisheries Cooperative Association, personal communication). Large amounts of *U. pinnatifida* sporophyll have been discarded depending on the market price, which fluctuates widely each year (Funato, Okirai Fisheries Cooperative Association, personal communication). In addition, the United Nations has adopted as its fourteenth sustainable development goal: “Conserve and sustainably use the oceans, seas and marine resources for sustainable development” [50]. Therefore, short-term culture of *M. nudus* collected from barrens using these discarded portions of cultivated kelps is a highly cost-effective strategy that could contribute to achieving the goals of higher performance of sustainable aquaculture.

## 5. Conclusions

A larger number of odor-active compounds were detected in gonads of BS and SU than those of NF. The pleasant “green”, “sour”, and “fruity” aromas from butyl acetate detected from gonads of BS and SU are here recognized as essential flavor components of high-quality gonads. The taste of sea urchin gonads analyzed by a taste-sensing system revealed a large variation depending upon the kind of food that sea urchins were fed. The gonads of BS and SU treatments had a strong umami taste compared to those of NF, WEB and WBA sea urchins. The taste-sensing system has a high potential to accurately measure the combination of amino acids and 5′-ribonucleotides content contributing to the umami taste of gonads. Gonads with “green” and “fruity” aromas from butyl acetate, “sweet” aroma from benzaldehyde, umami EIT > 13.8, bitterness EIT < 3.1, and without an unpleasant sulfur odor (low level of sulfur-containing compounds) can be highly marketable. A test omitting odor-active compounds is still required to clearly identify the compounds constituting the overall flavor of high-quality gonads.

## Figures and Tables

**Figure 1 sensors-20-07008-f001:**
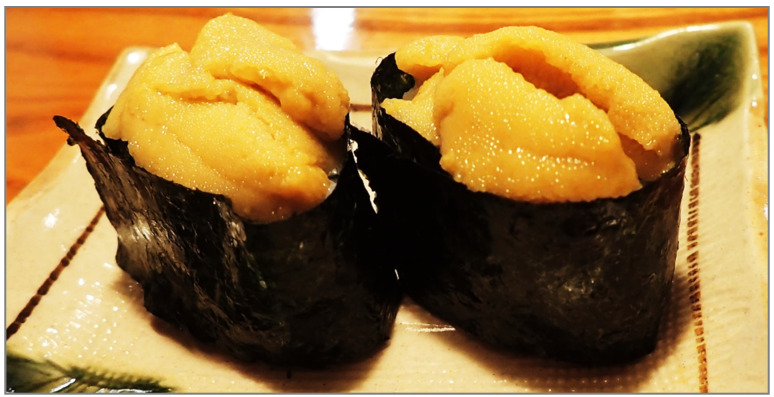
Sushi made with *Mesocentrotus nudus* gonads, which were evaluated as high-quality by a chef from a sushi restaurant with a Michelin 3-star rating in Tokyo (photo by S. Takagi).

**Figure 2 sensors-20-07008-f002:**
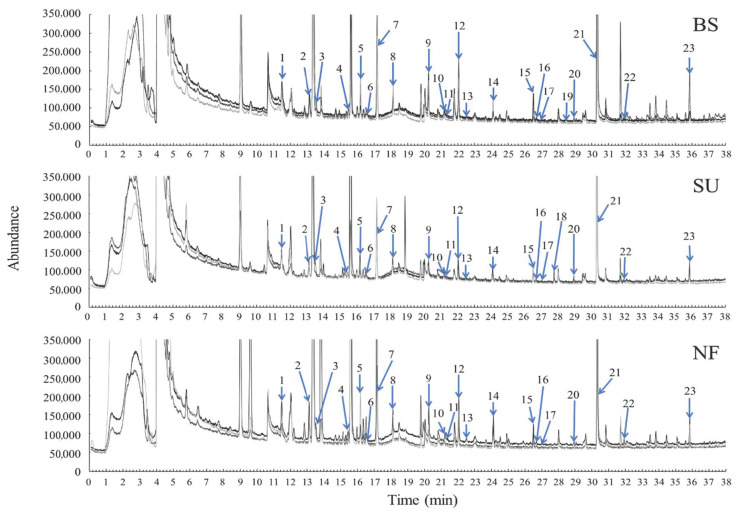
Chromatograms of odor-active organic compounds in the gonads of *Mesocentrotus nudus* using gas chromatography–mass spectrometry (N = 3). BS, SU and NF indicate sea urchins fed with the basal frond portions of *Saccharina japonica*, sporophylls of *Undaria pinnatifida* and no food, respectively. Compounds are identified by peak numbers, as shown in Table 1.

**Figure 3 sensors-20-07008-f003:**
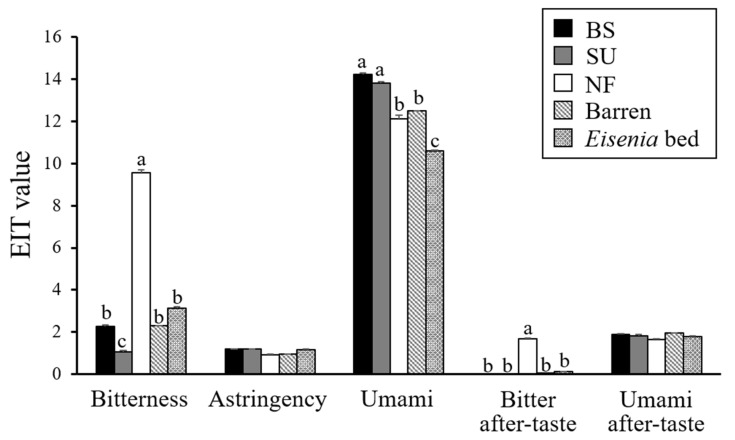
Estimated intensity of taste (EIT) of gonads of *Mesocentrotus nudus* fed experimentally or obtained directly from the wild. Experimentally fed: BS, fed with the basal frond portions of *Saccharina japonica*; SU, sporophylls of *Undaria pinnatifida*; NF, no food. Wild sea urchins: “Barren”, collected from a seabed barren; “Eisenia bed”, from an *Eisenia bicyclis* bed. Bars indicate mean EIT ± SE. Lower-case letters indicate more than one-unit difference in each EIT among groups, indicating a recognizable difference in taste by humans [30].

**Table 1 sensors-20-07008-t001:** Retention time, odor descriptions, detection and peak area of the odor-active volatile organic compounds detected in *Mesocentrotus nudus* gonads using gas chromatography (GC)-sniffing (SNF) and GC–mass spectrometry (GCMS).

SNF-RT	GCMS-RT	Compound	NO.	Description	SNF Detection			Peak Area			
					BS	SU	NF	BS	SU	NF	*p*
11.50	11.47	Ethyl acetate	1	Fresh, green *	+			186.07 ± 12.24	133.22 ± 27.31	154.65 ± 44.16	0.514
12.60–12.70		Unknown		Sea urchin *	+						
13.20	13.13	2-Propanol	2	Fresh, green, fishy ^†^		+	+	224.42 ± 27.05	168.95 ± 28.73	307.15 ± 59.18	0.134
13.50–13.60	13.52	Benzene	3	Sweet, caramel *	+	+		34.62 ± 9.12	42.31 ± 10.80	49.83 ± 17.04	0.716
15.48	15.45	Decane	4	Green *			+	83.07 ± 13.52	61.35 ± 5.68	69.84 ± 16.20	0.511
16.20–16.30	16.17	Chloroform	5	Putrid, fishy ^†^	+		+	238.68 ± 86.48	126.32 ± 49.58	287.84 ± 121.82	0.481
16.73–16.75	16.60	α-Pinene	6	Sea urchin ^†^		+	+	9.83 ± 2.01	5.35 ± 0.93	9.44 ± 1.92	0.198
17.18	17.18	Toluene	7	Green, chemical	+		+	933.53 ± 80.79	757.52 ±33.75	1695.73 ± 643.82	0.196
18.03–18.04	18.08	Butyl acetate	8	Fruit, sour, green *	+	+		8.16 ± 0.77	6.88 ± 1.27	8.82 ± 3.47	0.846
18.30		Unknown		Sea urchin *	+						
20.38–20.41	20.25	Xylene	9	Orange, fresh, sweet *		+		313.71 ± 51.19 ^a^	104.56 ±11.22 ^b^	126.15 ± 18.83 ^b^	0.004
21.30	21.23	Dodecane	10	Fishy ^†^			+	8.89 ± 0.79	7.05 ± 0.85	9.89 ± 1.24	0.198
	21.32	Ethyl toluene	11					9.61 ± 1.97	6.23 ± 1.03	6.72 ± 2.22	0.418
21.69		Unknown		Sea urchin *		+					
22.05	22.06	Limonene	12	Sea urchin ^†^	+			408.43 ± 78.32	153.03 ± 24.23	306.63 ± 59.68	0.060
22.61	22.52	Propyl benzene	13	Putrid, smoke ^†^		+		8.70 ± 0.33	6.42 ± 0.57	8.97 ± 1.82	0.284
22.78		Unknown		Fishy ^†^		+					
24.13	24.12	Styrene	14	Putrid, *Undaria* ^†^		+		30.38 ± 3.86 ^b^	18.57 ± 2.62 ^b^	60.43 ± 11.53 ^a^	0.005
26.63–26.66	26.50	6-Methyl-5-hepten-2-one	15	Sea urchin ^†^		+	+	113.72 ± 56.79	104.01 ± 19.40	117.70 ± 33.72	0.969
	26.67	Isopropyl benzene	16					9.03 ± 0.60	5.95 ± 0.40	8.69 ± 1.33	0.088
27.18	27.00	Mentha-1,4,8-triene	17	Sea urchin after taste	+			4.07 ± 0.19	3.11 ± 0.35	4.12 ± 0.41	0.123
27.30		Unknown		Sea urchin, citrus *			+				
27.85	27.79	3-Octanol	18	Sea urchin *		+		ND ^b^	59.66 ± 17.09 ^a^	ND ^b^	<0.001
28.50	28.47	2-Butoxyethanol	19	Green, heavy, seaweed	+			4.43 ± 1.35 ^a^	ND ^b^	ND ^b^	<0.001
29.00–29.04	28.95	Ethyl octanoate	20	Sea urchin, sour and sweet ^†^	+	+	+	8.28 ± 2.13	9.00 ± 3.73	8.16 ± 0.47	0.973
30.50	30.32	2-Ethylhexanol	21	Green *		+		5198.37 ± 3375.74	2396.09 ± 403.79	4171.60 ± 1338.62	0.731
31.87–31.89	31.92	Benzaldehyde	22	Sweet, floral, fresh *	+	+	+	8.60 ± 1.09	8.21 ± 0.81	13.01 ± 3.38	0.276
32.30		Unknown		Sea urchin, fresh shellfish *	+						
33.39–33.50		Unknown		Heavy, butter ^†^	+	+					
34.81		Unknown		Sea urchin *	+						
35.90	35.86	2-Methyl-6-methylene-2,7-octadienal	23	Sweet *		+		218.91 ± 125.88	122.85 ± 27.19	169.77 ± 59.04	0.885
36.60		Unknown		Sea urchin, fresh shellfish	+		+				

RT, retention time (min). Odor descriptions: *, pleasant; ^†^, unpleasant. For each compound, compound number (NO.), detection by SNF (+) and peak area values (×10^−4^; mean ± S.E.; n = 3) are provided for the three experimental feeding treatments: BS, basal frond portions of *Saccharina japonica*; SU, sporophylls of *Undaria pinnatifida*; NF, no food. Significance of values in peak areas for each compound among treatments determined by ANOVA are provided (*p*). Superscript letters “a” and “b” indicate significant differences in peak areas among treatments by Tukey’s test (*p* < 0.05 by Tukey’s test). ND, not detected.

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
