# Peer review of "Quantification of the Flavor and Taste of Gonads from the Sea Urchin Mesocentrotus nudus Using GC–MS and a Taste-Sensing System"

_sensors, 2020, doi:10.3390/s20247008_

Round 1

Reviewer 1 Report

The manuscript entitled "Quantification of the flavor and taste of gonads from the sea urchin Mesocentrotus nudus using GC-MS and a taste-sensing system" can be accepted by SENSORS after major revision.

  1. The difference of gas composition of sea urchin is helpful to detect the flavor of sea urchin. But will other similar organic gases interfere with the detection?
  2. Adding some sensing related work.
  3. Adding a nice ABSTRACT GRAPH.
  4. How about the detection speed? Is the velocity related to the concentration of the object to be measured?

Author Response

The manuscript entitled "Quantification of the flavor and taste of gonads from the sea urchin Mesocentrotus nudus using GC-MS and a taste-sensing system" can be accepted by SENSORS after major revision.

  1. The difference of gas composition of sea urchin is helpful to detect the flavor of sea urchin. But will other similar organic gases interfere with the detection?

Thank you for your comments. As we mentioned in supplementary materials, the gonads were incubated in a glass jar and the headspace gas was vacuum-extracted from the glass jar. Thus, the organic gases we detected were derived from gonads.

We added the detail of methods of GC-MS analysis which were mentioned in supplementary materials on the lines 122–135 and 143–149 in 2.2. Odor-active volatile compound analysis.

  1. Adding some sensing related work.

Thank you for your advice. We added discussion of the potential of the taste sensing system referring sensing works using the taste sensing system for seafood evaluation. We added the sentences “Kitaoka et al. [18] suggested that suggested the umami sensor would reflect synergistic effects of glutamic acid, IMP and AMP by the taste evaluation of pacific oyster C. gigas collected from three different localities” on the lines 284–286, and “Past studies showed the potential of the taste sensing system to evaluate taste of C. gigas [18], the swimming crab P. trituberculatus [19], the bluefin tuna T. orientalis, the yellowtail S. quinqueradiata, and the squids S. oualaniensis and T. pacificus [16].” on the lines 292–295 in 4.2. Taste analysis. In addition, we changed the sentence “Therefore, the taste-sensing system has a high potential to reliably and economically evaluate the umami taste of sea urchin gonads” to “Therefore, we conclude the taste-sensing system has a high potential to reliably and economically evaluate the umami taste of sea urchin gonads” on the lines 295–396 in 4.2. Taste analysis.

  1. Adding a nice ABSTRACT GRAPH.

We made graphical abstract.

We hope it could be acceptable for you.

  1. How about the detection speed? Is the velocity related to the concentration of the object to be measured?

Thank you for the comment. As you mentioned, the velocity can be related to the concentration of the object. In the present study, we set the helium carrier gas at mean linear velocity of 20.6 cm sec-1 same as past studies (Sato et al. 2019, Food and Nutrition Sciences 10, 860–875; Takagi et al. 2020, Plos One 15, e0231673) to be comparable to them. We mentioned on the lines 135–137 in 2.2. Odor-active volatile compound analysis.

We deeply appreciated you valuable, insightful comments.

Reviewer 2 Report

The article by Agatsuma et el. titled "Quantification of the flavor and taste of gonads from the sea urchin Mesocentrotus nudus using GC-MS and a taste-sensing system" investigates the difference in flavor and taste of different sea urchin gonads by gas chromatography-mass spectrometry and taste sensing system.

The articles can be published after the following comments are addressed-

1. Line 34-40 in the introduction section must be rewritten. For instance: Fresh sea urchin gonad is an important item of Japanese “sushi” cuisine, Tokyo Metropolitan Central Wholesale Market is the largest wholesale sea urchin market in the world [2] and the average price of sea urchin gonads there doubled from 7,306 JPY.kg-1 in 2008 to 14,683 JPY.kg-1 in 2018 [3] 

The italicized sentence is too long, confusing and grammatically incorrect. It creates bad impression to the readers about the article, especially when first few lines of the introduction seem incorrect.

2. The gender of the sea urchins were never mentioned. How important is it for the production of sensory compounds?

3. The author should briefly report sampling method in the main article (not only in supplement).

4. The GC column type and dimension for analysis must be mentioned in main article.

5. Why there is no representative TIC chromatogram in the article?

Author Response

The article by Agatsuma et el. titled "Quantification of the flavor and taste of gonads from the sea urchin Mesocentrotus nudus using GC-MS and a taste-sensing system" investigates the difference in flavor and taste of different sea urchin gonads by gas chromatography-mass spectrometry and taste sensing system.

The articles can be published after the following comments are addressed.

  1. Line 34-40 in the introduction section must be rewritten. For instance: Fresh sea urchin gonad is an important item of Japanese “sushi” cuisine, Tokyo Metropolitan Central Wholesale Market is the largest wholesale sea urchin market in the world [2] and the average price of sea urchin gonads there doubled from 7,306 JPY.kg-1 in 2008 to 14,683 JPY.kg-1 in 2018 [3] 

The italicized sentence is too long, confusing and grammatically incorrect. It creates bad impression to the readers about the article, especially when first few lines of the introduction seem incorrect.

We appreciate your insightful comments. We changed the sentence to “Fresh sea urchin gonad is an important item of Japanese “sushi” cuisine. Tokyo Metropolitan Central Wholesale Market is the largest wholesale sea urchin market in the world [2], where the average price of sea urchin gonads doubled from 7,306 JPY.kg-1 in 2008 to 14,683 JPY.kg-1 in 2018 [3].

  1. The gender of the sea urchins were never mentioned. How important is it for the production of sensory compounds?

Thank you for your comment. Past studies reported sexual difference in flavor and odor of gonads from the sea urchin Paracentrotus lividus (Baião et al. 2020, LWT-Food Sci Tech 130, 109629) and Evechinus chloroticus (Niimi et al. 2010, Food Chem 121, 601607; Phillips et al. 2010, LWT-Food Sci Technol 43, 202–213). However, sea urchin gonads are traded without sex separation in the market because we could not identify the sex of unmatured sea urchin without histological observation. In the present study, because we had to conduct GC-MS and GC-sniffing analyses immediately after dissection to analyze fresh gonads, histological observation could not be conducted. Furthermore, the gonads of sea urchins fed no food (NF) and from the barren (WBA) were too small to conduct the analyses individually, and some individuals of each group have to be combined for the analyses (therefore we collected gonads from 10–15 individuals of each group with three replication). Thus, it was impossible to separate sex in the present study. In addition, the purpose of the present study is to standardize flavor and taste of high-quality gonads. Thus, we think we do not need to discuss about sexual difference in the present study.

  1. The author should briefly report sampling method in the main article (not only in supplement).

Certainly. We should mention the detail of sampling methods in the main article.

We added the sentences “Headspace volatile compounds were collected in a large-volume static headspace (LVSH) system (Entech 7100A series, Entech Instruments Inc., Simi Valley, CA, USA). Each gonad from a container was sealed in a 375 ml glass jar for measurement of LVSH and stored in an incubator (DK400, Yamato Scientific Co., Ltd., Tokyo, Japan) at 30 ºC for 10 min. After incubation, 150 ml of headspace gas was vacuum-extracted from the glass jar. The volatile organic compounds (VOCs) were desorbed by thermos desorption using a pre-concentrator (Entech 7100A series, Entech Instruments Inc., Simi Valley, CA, USA) and applied to the GC-MS system.” on the lines 122129, “Quantification of the volatile compounds was performed using an Agilent 6890 series gas chromatograph (Agilent Technologies Inc., Palo Alto, CA, USA) equipped with an Agilent 5975B mass-selective detector and a sniffing port. One half of the column flow was directed to the MS system, while the other half was directed to the heated sniffing port. The GC-MS system was equipped with a DB-WAX column (60 m × 0.25 mm i.d., 0.5 µm film thickness; 122-7063, Agilent Technologies Inc.).” on the lines 130–135, “One half of the column flow was directed to a heated sniffing port (ODP2 Olfactory Detection Port, Gerstel GmbH & Co.KG, Mülheim an der Ruhr, Germany) for GC-sniffing analysis. Humidified air (50–75% relative humidity) was carried to the sniffing port at 1.02 ml min-1. The panelists, who are well versed in sea urchin gonad quality and share common perceptions, recorded the retention time and the related description of the aroma compounds [28, 29] by writing in a paper. The relative amounts of each volatile compound detected by GC-sniffing analysis was calculated based on the peak areas in the chromatograms.” on the lines 143–149 in 2.2. Odor-active volatile compound analysis, and “Approximately 10 g gonads of each group were finely cut into a paste with scissors, and diluted with nine volumes of distilled water. The mixture was boiled for 15 min and then filtered through a nylon net (Draining bag; 6-0709-0601, Daicel FineChem Ltd., Tokyo, Japan) to obtain the sample extract. The filtrates were estimated for taste intensity with a taste sensing system (SA402B, Intelligent Sensor Technology. Inc., Kanagawa, Japan) according to the manufacturer’s protocol.” on the lines 153–158 in 2.3. Taste analysis.

According to the addition, we deleted the supplementary materials, and added the references McDonnell et al. (2001) and Phillips et al. (2009) to the reference list and changed the reference numbers after NO. 28. In addition, we changed the sentences “Odor-active volatile compounds in the gonad tissues were analyzed by GC-MS and GC-sniffing techniques within 48 hours after dissection, according to Sato et al. [26] (see supplementary information in detail)” to “ Odor-active volatile compounds in the gonad tissues were analyzed by GC-MS and GC-sniffing techniques within 48 hours after dissection, according to Sato et al. [26]” on the lines 121–122 in 2.2. Odor-active volatile compound analysis, and “The estimated intensity of taste (EIT) of sea urchin gonads was analyzed according to the method of Touhata et al. [16] with a slight modification (see supplementary information in detail)” to “The estimated intensity of taste (EIT) of sea urchin gonads was analyzed according to the method of Touhata et al. [16] with a slight modification” on lines 151–152 in 2.3. Taste analysis.

  1. The GC column type and dimension for analysis must be mentioned in main article.

As we mentioned in Q3, we added the inform of the GC column type (DB-WAX column (60 m × 0.25 mm i.d., 0.5 µm film thickness; 122-7063, Agilent Technologies Inc.)) on the lines 133–135 in 2.2. Odor-active volatile compound analysis. We appreciate your comment.

  1. Why there is no representative TIC chromatogram in the article?

Thank you for your comment. We added the TIC chromatogram as Figure 2.                                                                                                  

In addition, we added compound NO. in the Table 1, and changed former Figure 2 to Figure 3.

We deeply appreciate your valuable comments.

Other changes

The current addresses of Yuko Murata, Ken Touhata and Noriko Ishida changed.

Thus, we added the information in the manuscript.